# Effects of exogenous melatonin on sugar and organic acid metabolism in early-ripening peach fruits

**Kexuan Zhou**[1☯], **Qi Cheng**[1☯], **Jingtong Dai**[1], **Yuan Liu**[1], **Qin Liu**[1], **Rui Li**[1], **Jiangyue Wang**[1], **Rongping Hu**[2], **Lijin Lin**[1]*

**1** College of Horticulture, Sichuan Agricultural University, Chengdu, Sichuan, China, **2** Institute of Agricultural Resources and Environment, Sichuan Academy of Agricultural Sciences, Chengdu, Sichuan, China

☯ These authors contributed equally to this work.
* llj800924@qq.com

**Data Availability Statement:** All relevant data are within the paper and its Supporting Information files.

## Abstract

To evaluated the effects melatonin (MT) on the sugar and acid metabolism of early-ripening peach fruits, the concentration of 150 μmol/L MT was sprayed on the leaves of peach trees. MT increased the contents of total soluble sugar and sucrose in peach fruits during the whole ripening period, and increased the contents of glucose and sorbitol at the mature stage. During the whole ripening period, MT also increased the activities of sucrose synthase, sucrose phosphate synthase, neutral invertase, and acidic invertase and the relative expression levels of *sucrose synthase*, *sucrose phosphate synthase*, *neutral invertase*, and *acidic invertase* genes, while decreased the activity of sorbitol oxidase and the relative expression level of *sorbitol dehydrogenase* to some extent. Moreover, MT decreased the contents of total organic acid, malic acid, and citric acid at mature stage. At mature stage, MT decreased the activities of citrate synthetase and phosphoenolpyruvate carboxylase and the relative expression levels of *citrate synthetase* and *phosphoenolpyruvate carboxylase* genes, while increased the relative expression levels of *Nicotinamide adenine dinucleotide phosphate* (*NADP⁺*)-*malic enzyme*, *malate dehydrogenase*, and *aconitase* genes. Therefore, MT promotes the sugar accumulation and organic acid degradation in early-ripening peach fruits.

## Introduction

Peach (*Prunus persica* L.) is a widely grown fruit worldwide with high nutritional value and tasteful flavor [1]. China is the world's largest producer of peaches, with the production area of early-ripening peaches being more than 70% [2]. The main soluble sugars in peach fruits are sucrose, glucose, fructose, and sorbitol [3], while the main organic acids are malic acid, citric acid, and quinic acid [4]. These soluble sugars are responsible for the peach fruit sweetness upon ripening [4, 5], and their metabolism is controlled by the different enzymes, including sucrose synthase (SS), sucrose phosphate synthase (SPS), neutral invertase (NI), acidic invertase (AI), sorbitol dehydrogenase (SDH), and sorbitol oxidase (SOX) [6]. Conversely, the

**Funding:** YES - This work was financially supported by the College Student Innovation Training Program Project of Sichuan Agricultural University (202110626079). The funders had roles in study design, data collection and analysis, decision to publish, and preparation of the manuscript.

**Competing interests:** The authors have declared that no competing interests exist.

organic acids determine the flavor quality of peach fruits [5, 7]. The malic enzyme (ME), malate dehydrogenase (MDH), Nicotinamide adenine dinucleotide phosphate (NADP$^+$)-malic enzyme (NADPME), and phosphoenolpyruvate carboxylase (PEPC) regulate malic acid metabolism [8], while citrate synthetase (CS) and aconitase (ACO) regulate citric acid accumulation [9]. However, the low sugar content and sugar/acid ratio in the early-ripening peaches reduces sweetness resulting in poor flavor [10, 11]. Therefore, improving the sugar content and regulating the sugar/acid ratio of the early-ripening peach fruits are important.

Plant growth regulators such as nitric oxide [12, 13], brassinolides [14], methyl jasmonate, and propyl dihydrojasmonate [15] have been used to improve peach fruit quality. However, residual problems are associated with these plant growth regulators [16]. Melatonin (MT) is a natural endogenous plant hormone widely present in crops [17–19] and has been employed to improve the resistance of crops to adverse conditions [20–22]. MT treatment improved the fruit quality of cherry tomatoes by increasing their soluble sugar content and regulating the sugar/acid ratio [23]. MT treatment also promoted the contents of total soluble sugars, soluble solids, and vitamin C in watermelon fruits [24]. For fruit trees, MT treatment promoted sucrose and fructose accumulation in grape leaves by stimulating the sucrose metabolism-related enzymes [25]. The treatment also increased the solid soluble content and decreased the titratable acid content in grape berries [26]. Furthermore, MT treatment increased the sugar content in pear fruit during development by promoting the accumulation of early starch content, regulating the invertase, SS, and SPS activities, and increasing the sucrose and sorbitol contents in pear fruits [27]. MT treatment also inhibited the AI and NI activities and increased the SS and SPS activities to promote sugar accumulation in pear fruits [28]. Thus, these studies show that MT can be used to improve the quality of fruits.

Most previous studies on the use of MT in peach fruits mainly focused on the postharvest effects of the treatment [29–31]; however, the effects of MT on the sugar and acid metabolism during peach fruit development have not been reported. In our previous study, the MT treatment improved the fruit quality of early-ripening peaches, with the best MT concentration being 150 μmol/L [32]. Therefore, this study evaluated the effects of 150 μmol/L MT on the sugar and acid metabolism of early-ripening peach fruits. This study aimed to establish a viable method for improving the sweetness and flavor of early-ripening peaches and provide the theoretical basis for their MT-mediated quality improvement.

## Materials and methods

### Materials

The peach variety used in this experiment was 'Zaomi' grafted on wild peach rootstock. The grafted peach seedlings were planted in the orchard at Chengdu Academy of Agricultural and Forestry Sciences, Chengdu, China, in February 2015. The row spacing was 4 m, and the plant spacing was 3.5m. The cultivation method was the high monopoly, with a monopoly width and height of 2 m and 0.5 m, respectively, and a furrow width of 0.5 m. MT was obtained from Beijing Solarbio Science & Technology Co., Ltd (Beijing, China).

### Experimental design

Seven-year-old 18 peach trees were selected for the experiment in May 2022, when the peach fruits were at a rapid expansion period (45 days after flowering). The selected peach fruits were bagged in white double-shade bags and divided into two groups, each containing 9 peach trees. Thereafter, MT solution (150 μmol/L) [32] was sprayed on the leaves of the trees, with 1.5 L of the solution per tree. The control (CK) was sprayed with the same volume of water. Therefore, there were two treatments and each was conducted in triplicate, with three trees per

replicate. The MT and water treatments were sprayed every 7 days with four times in total. The first fruit collection was conducted at 0 days after treatment, after which the fruits were collected every 7 days (two fruits from each side of each tree) until maturity, totaling five times. The fruit collection at each time was conducted before spray application. After collection, the fruits were placed in an ice box and taken to the laboratory, where they were cut open to remove the cores and stalks, and the fruit pulps were immediately stored at -80˚C.

## Sugar content determination

The contents of total soluble sugar, sucrose, glucose, sorbitol, and fructose were determined using high-performance liquid chromatography (Agilent 1260 HPLC system, Agilent Technologies). The chromatographic column was Innoval $NH_2$ (250 mm × 4.6 mm, 5 μm, Agela Technologies, Shanghai, China), and acetonitrile-water (80:20, V:V) was used as the mobile phase. The analysis involved a differential detector with a flow rate of 1 mL/min, a detection cell temperature of 40˚C, a column temperature of 35˚C, and an injection volume of 10 μL [33].

## Determination of organic acid content

The contents of total acid, malic acid, and citric acid were also determined using high-performance liquid chromatography (Agilent 1260 HPLC system, Agilent Technologies). The chromatographic column was MZ PerfectSil Target C18 (250 mm × 4.6 mm, 5 μm,), with 4% aqueous methanol as the mobile phase. The analysis involved a flow rate of 0.5 mL/min, a column temperature of 25˚C, a detection wavelength of 210 nm, and an injection volume of 10 μL [34, 35].

## Determination of the activities of sugar and organic acid metabolism-related enzymes

The activities of sugar and organic acid metabolism-related enzymes (SS, SPS, NI, AI, SDH, SOX, ME, MDH, CS, PEPC, and ACO) were determined using the enzyme-linked immunosorbent assay (ELISA) kits (Shanghai Enzyme Link Biotechnology Co., Ltd., Shanghai, China), according to the manufacturer's instructions.

## Determination of sugar and organic acid metabolism-related gene expression levels

Total RNA was extracted using an M5 Plant RNeasy Complex Mini Kit (Mei5 Biotechnology Co., Ltd., Beijing, China), according to the manufacturer's instructions. Thereafter, the MF166-plus-M5 Super qPCR RT kit with gDNA remover (Mei5 Biotechnology Co., Ltd., Beijing, China) was used for reverse transcription. All primer sequences (Table 1) were designed by the Primer 5.0 software using the peach cDNA sequences published in the NCBI. The housekeeping gene was the *PpActin* (Table 1) [36]. Primer synthesis was conducted by the Chengdu Branch of Beijing Tsingke Biotechnology Co., Ltd. (Chengdu, China). Moreover, quantitative PCR was conducted on a real-time quantitative PCR instrument (CFX Connect; Bio-Rad, Hercules, CA, USA) using 2X M5 HiPer SYBR Premix EsTaq (with Tli RNaseH) (Mei5 Biotechnology Co., Ltd.). The amplification was performed in triplicate according to the standard protocol of the ABI StepOnePlus Real-Time PCR System (Applied Biosystems, CA, USA). The $2^{-\Delta\Delta CT}$ method was used to calculate the relative levels of expression.

**Statistical analysis.** The data were analyzed in triplicate using the SPSS 20.0.0 software (IBM, Chicago, IL, USA) and subjected to the Student's t-test ($0.01 \leq p < 0.05$ or $p < 0.01$). The graphs were generated using Microsoft Excel 2010.

**Table 1. Primer information for the quantitative real-time PCR.**

| Gene ID in NCBI | Gene name | Gene description | Sequence 5'–3' |
|---|---|---|---|
| LOC18779708 | *PpActin* | *Actin* | F: AACTGGAATGGTGAAGGCTGG |
| | | | R: GGGCTTCATCACCTACATAGGC |
| LOC18770262 | *PpSS* | *SS* | F: TTGTGATCCTTTCTCCCCACG |
| | | | R: CAGTCCCTGTTGCTTAATACGC |
| LOC18790710 | *PpSPS* | *SPS* | F: GTTCCTCCAGAGAGCCCCTT |
| | | | R: ACCACAACTCACATTCCAGCA |
| LOC18788228 | *PpNI* | *NI* | F: TTCACCGCAGCCGAAATTG |
| | | | R: GTCGTCTTGCAAACCCTAGTC |
| LOC18784736 | *PpAI* | *AI* | F: TTGGAGACGTCCGCTAATGG |
| | | | R: CGGGTCATCAGGTATCCACG |
| LOC18787602 | *PpSDH* | *SDH* | F: GAAAACATGGCTGCTTGGCT |
| | | | R: CATCACTTCCGCATATCCCAAC |
| LOC18779761 | *PpNADPME1* | *NADPME* | F: GCGTAGTGATGGAGAGCACA |
| | | | R: CTCGGTGGCAGTATCCTCAC |
| LOC18788358 | *PpMDH1* | *MDH* | F: ACCAATGATCGCAAGGGGAGT |
| | | | R: GGAAAGCAGCATCGACCAACT |
| LOC110755472 | *PpCS1* | *CS* | F: TTGTGAAGCAGCGTCTTGGC |
| | | | R: AAGCTCAATGTCGATCCAAGCC |
| LOC18790559 | *PpPEPC1* | *PEPC* | F: CTCTCCCTACATTCTGGCTGCA |
| | | | R: TGCACTTGGCATAGCAACCG |
| NM_001299190.1 | *PpACO* | *ACO* | F: CTTATGGAAGTCGCCGTGGT |
| | | | R: GCAGCATCAAACACGGACAG |

## Results

### Sugar contents in peach fruits

The contents of total soluble sugar and sucrose increased with the number of days after treatment (Fig 1A and 1B). Thus, MT treatment significantly increased the contents of total soluble sugar and sucrose during the whole ripening period. At 28 days after treatment, the MT treatment increased the total soluble sugar and sucrose contents by 18.01% and 18.88%, respectively, compared with the control. The contents of glucose and sorbitol increased when the days after treatment were less than 7 days but exhibited a decreasing trend after 7 days (Fig 1C and 1D). Compared with the control, MT treatment significantly increased the glucose content by 14.52% at 28 days after treatment and increased the sorbitol content at 7 and 28 days after treatment by 41.06% and 43.71%, respectively. However, the fructose content decreased with increasing days after the treatment (Fig 1E). Compared with the control, MT treatment significantly increased the fructose content 14 days after treatment.

### Sugar metabolism-related enzyme activities in peach fruits

The activities of sugar metabolism-related enzymes (SS, SPS, NI, AI, SDH, and SOX) had an increasing trend initially but later decreased with the increasing number of days after the treatment (Fig 2). During the whole ripening period, MT treatment significantly increased the activities of SS, SPS, and NI (Fig 2A–2C). Moreover, MT treatment increased the activities of SS, SPS, and NI by 32.39%, 95.66%, and 52.53%, respectively, at 28 days after treatment compared with the controls. MT treatment also increased the AI activity at 7, 14, and 21 days after treatment compared with the control (Fig 2D). However, MT had no significant effects on the

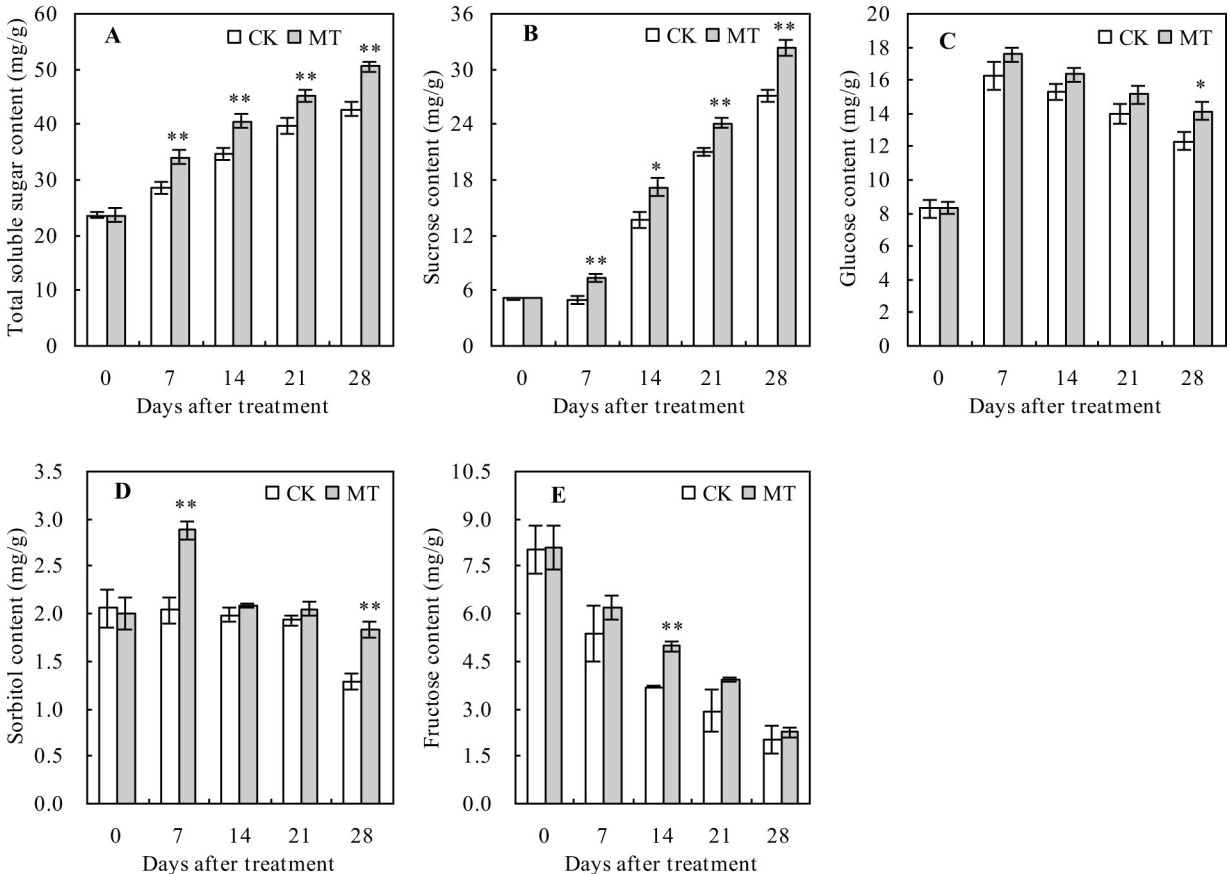

**Fig 1. Sugar contents in peach fruits.** Values represent the mean ± SD (n = 3). Asterisks indicate significant differences between the treatments using the Student's t-test (*: $0.01 \leq p < 0.05$; **: $p < 0.01$). **A**: total soluble sugar content; **B**: sucrose content; **C**: glucose content; **D**: sorbitol content; **E**: fructose content.

SDH activity at 7, 14, and 28 days after the treatment but significantly decreased the SDH activity at 21 days after the treatment, compared with the control (Fig 2E). Similarly, MT treatment significantly decreased the SOX activity by 57.52%, 50.16%, 53.57%, and 60.07% at 7, 14, 21, and 28 days after treatment, respectively, compared with the control (Fig 2F).

### Relative expression levels of sugar metabolism-related genes in peach fruits

The relative expression levels of sugar metabolism-related genes (*SS*, *SPS*, *NI*, *AI*, and *SDH*) initially had an increasing trend, which later decreased with the increasing number of days after the treatment (Fig 3). MT treatment significantly increased the relative expression levels of *SS*, *SPS*, and *NI* during the whole ripening period (Fig 3A–3C). Similarly, the relative expression level of *AI* was significantly increased by MT treatment at 14, 21, and 28 days after the treatment (Fig 3D), while that of *SDH* was significantly increased by MT treatment at 7, 14, and 21 days after the treatment (Fig 3E).

### Organic acid contents in peach fruits

The contents of total organic acid and citric acid had a decreasing trend (Fig 4A and 4C), while that of malic acid increased at first but decreased later with the increasing days after treatment (Fig 4B). Compared with the control, MT treatment significantly decreased the total

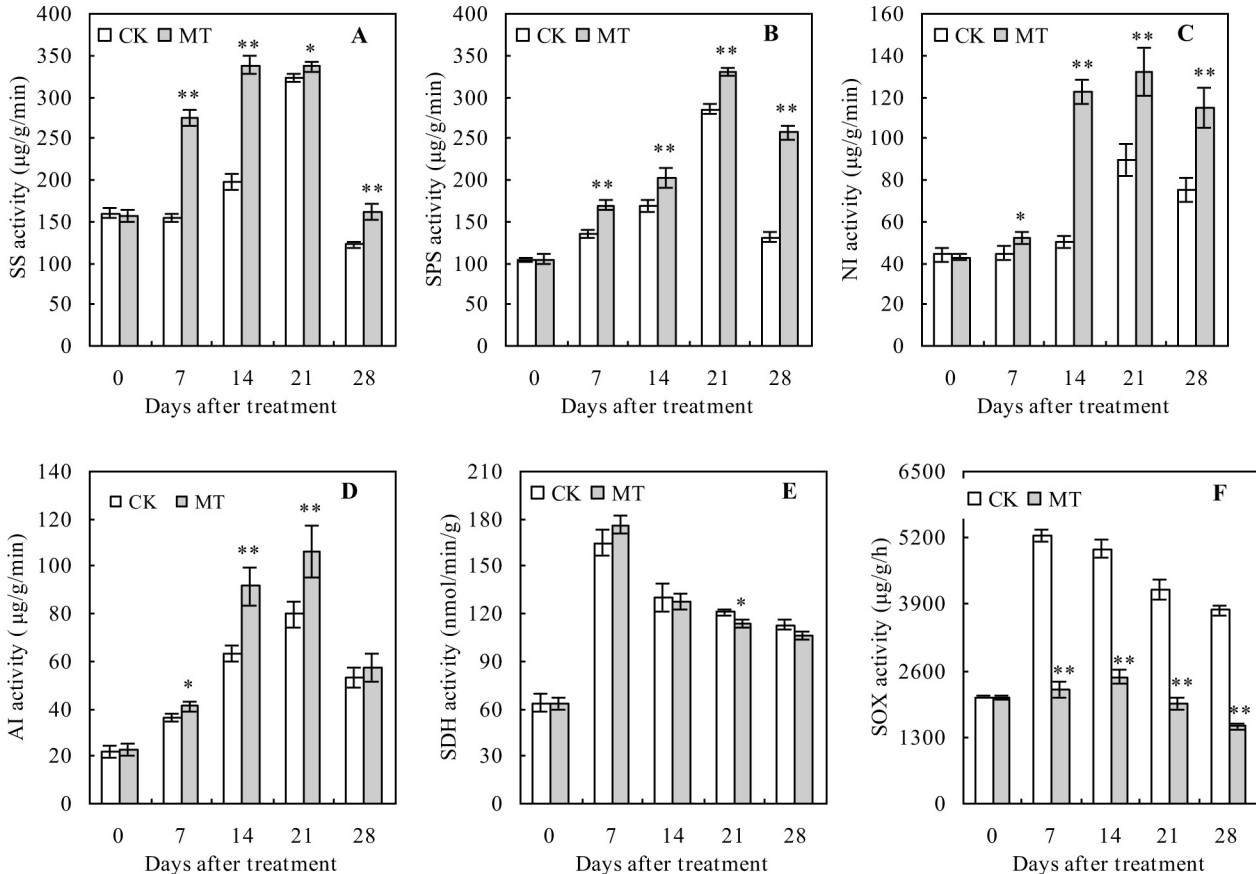

**Fig 2. Activities of the sugar metabolism-related enzymes in peach fruits.** Values represent the mean ± SD (n = 3). Asterisks indicate significant differences between the treatments using the Student's t-test (*: $0.01 \leq p < 0.05$; **: $p < 0.01$). **A**: sucrose synthase (SS) activity; **B**: sucrose phosphate synthase (SPS) activity; **C**: neutral invertase (NI) activity; **D**: acidic invertase (AI) activity; **E**: sorbitol dehydrogenase (SDH) activity; **F**: sorbitol oxidase (SOX) activity.

organic acid content 28 days after treatment but reduced the malic acid content 21 and 28 days after the treatment. Moreover, citric acid content was reduced by MT at 14 and 28 days after the treatment. The contents of total organic acid, malic acid, and citric acid reduced by 9.79%, 9.21%, and 16.72%, respectively, at 28 days after MT treatment compared with the control.

## Activities of the organic acid metabolism-related enzymes in peach fruits

The ME activity had an increasing trend with the increasing number of days after treatment but was significantly decreased by MT treatment at 7 days after the treatment compared with the control (Fig 5A). Compared with the control, MT treatment significantly decreased the MDH activity at 7 and 14 days after treatment but increased the activity at 28 days after treatment (Fig 5B). Moreover, the CS, PEPC, and ACO activities increased initially and then decreased with the increasing number of days after the treatment (Fig 5C–5E). MT treatment significantly increased the CS activity at 7 and 14 days after treatment but decreased the activity at 21 and 28 days after the treatment compared with the control (Fig 5C). PEPC activity was significantly increased at 7 days after treatment but decreased at 21 and 28 days after treatment with MT compared with the control (Fig 5D). However, ACO activity was significantly decreased by MT treatment at 7 days after the treatment compared with the control (Fig 5E).

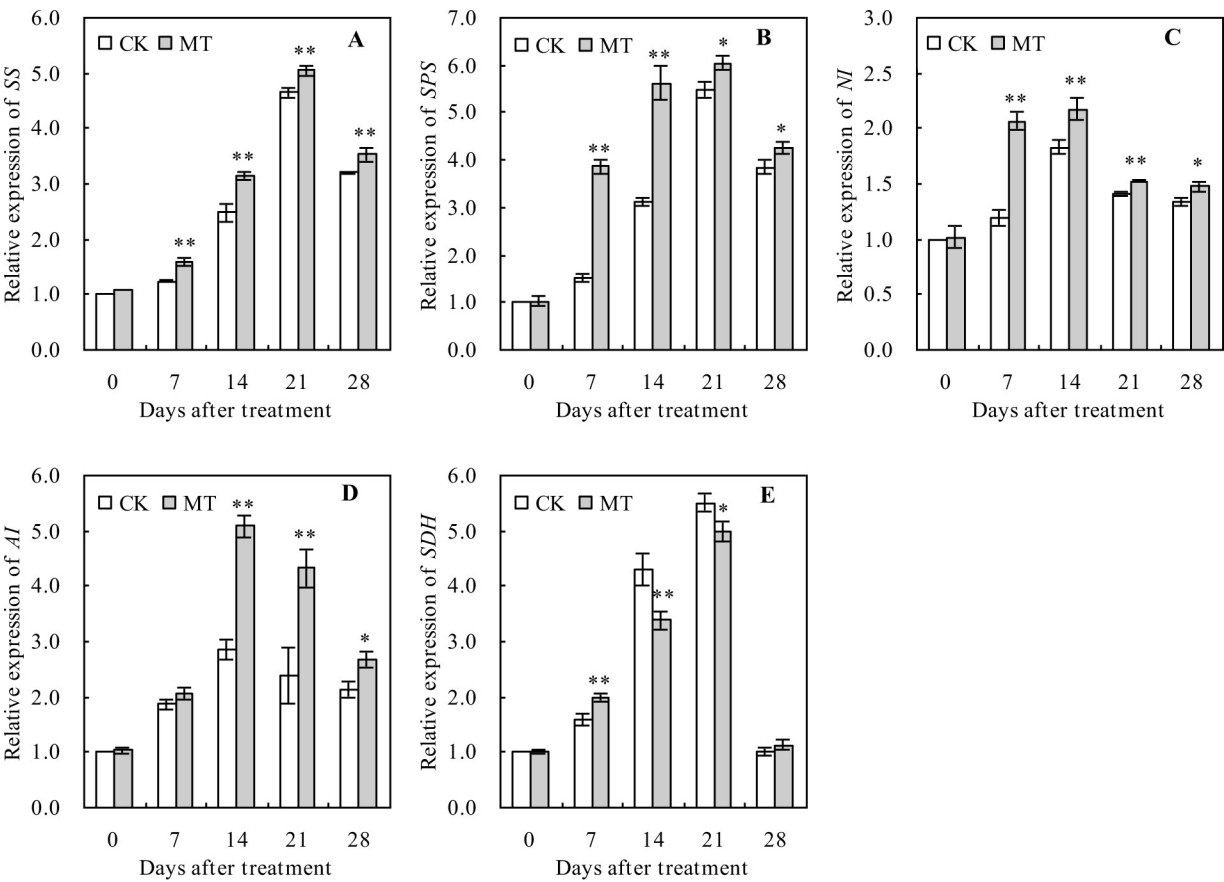

**Fig 3. Relative expression levels of sugar metabolism-related genes in peach fruits.** Values represent the mean ± SD (n = 3). Asterisks indicate significant differences between the treatments using the Student's t-test (*: $0.01 \leq p < 0.05$; **: $p < 0.01$). Relative expression levels of **A:** *sucrose synthase* (*SS*), **B**: *sucrose phosphate synthase* (*SPS*), **C**: *neutral invertase* (*NI*), **D**: *acidic invertase* (*AI*), and **E**: *sorbitol dehydrogenase* (*SDH*) genes.

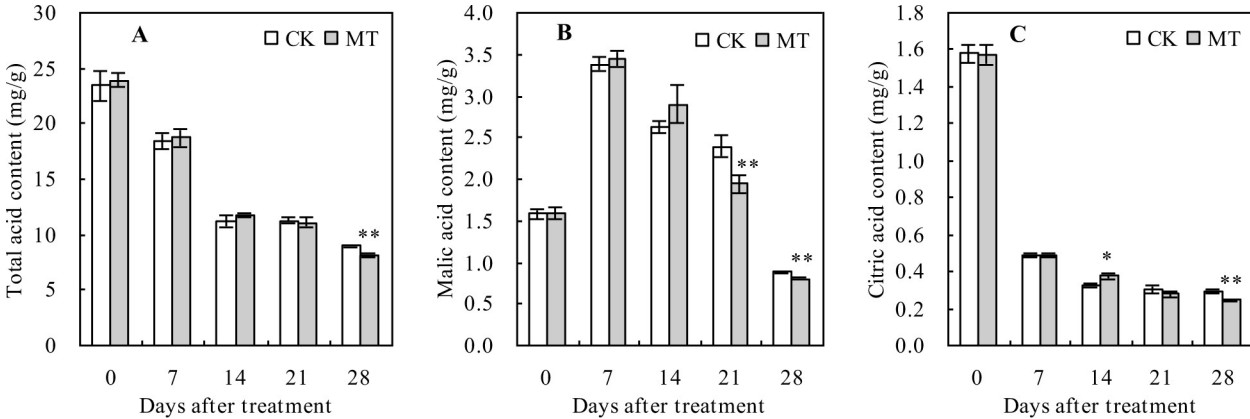

**Fig 4. Organic acid contents in peach fruits.** Values represent the mean ± SD (n = 3). Asterisks indicate significant differences between the treatments using the Student's t-test (*: $0.01 \leq p < 0.05$; **: $p < 0.01$). **A**: total acid content; **B**: malic acid content; **C**: citric acid content.

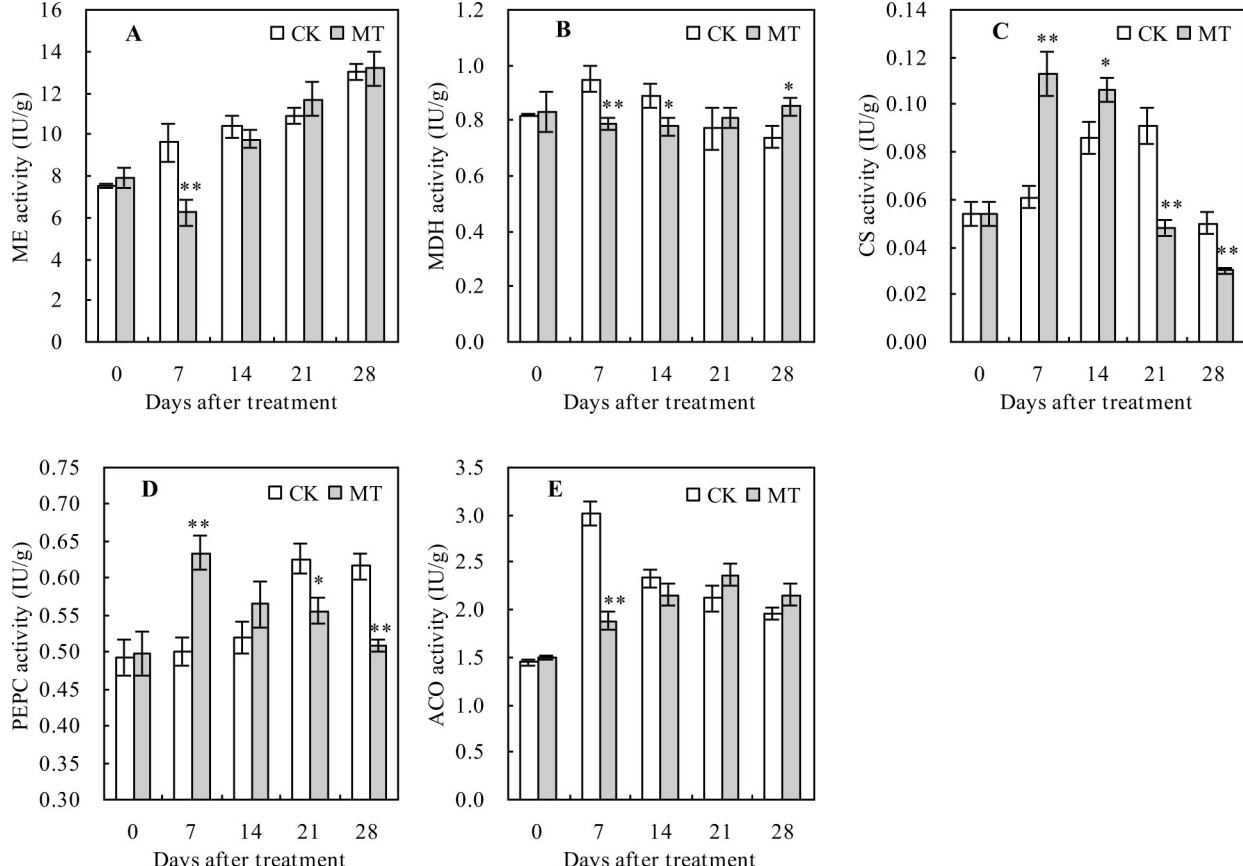

**Fig 5. Activities of the organic acid metabolism-related enzymes in peach fruits.** Values represent the mean ± SD (n = 3). Asterisks indicate significant differences between the treatments using the Student's t-test (*: $0.01 \leq p < 0.05$; **: $p < 0.01$). **A**: Malic enzyme (ME) activity; **B**: malate dehydrogenase (MDH) activity; **C**: citrate synthetase (CS) activity; **D**: phosphoenolpyruvate carboxylase (PEPC) activity; **E**: aconitase (ACO) activity.

## Relative expression levels of organic acid metabolism-related genes in peach fruits

MT treatment significantly decreased the relative expression levels of *NADPME*, *MDH*, and *ACO* at 7 and 14 days but increased these levels 21 and 28 days after the treatment, compared with the control (Fig 6A, 6B and 6E). However, MT treatment significantly increased the relative expression levels of *CS* and *PEPC* at 7 and 14 days but decreased these expression levels at 21 and 28 days after the treatment, compared with the control (Fig 6C and 6D).

## Discussion

Various enzymes are involved in sugar metabolism, among which SS and SPS regulate monosaccharide synthesis, while AI converts sucrose to glucose and fructose, and SDH and SOX catalyze sorbitol to glucose and fructose [37]. In the previous studies, the activities of SS and SPS gradually increased with the increasing number of days to peach fruit ripening, but the SS activity decreased prior to peach fruit ripening to promote sucrose accumulation. The AI activity was lower prior to peach fruit ripening but higher at the active ripening stage [38]. MT treatment increased the SPS activity but inhibited the invertase activity in pear fruit, thus promoting the rapid accumulation of sucrose [28]. MT treatment also decreased AI and NI

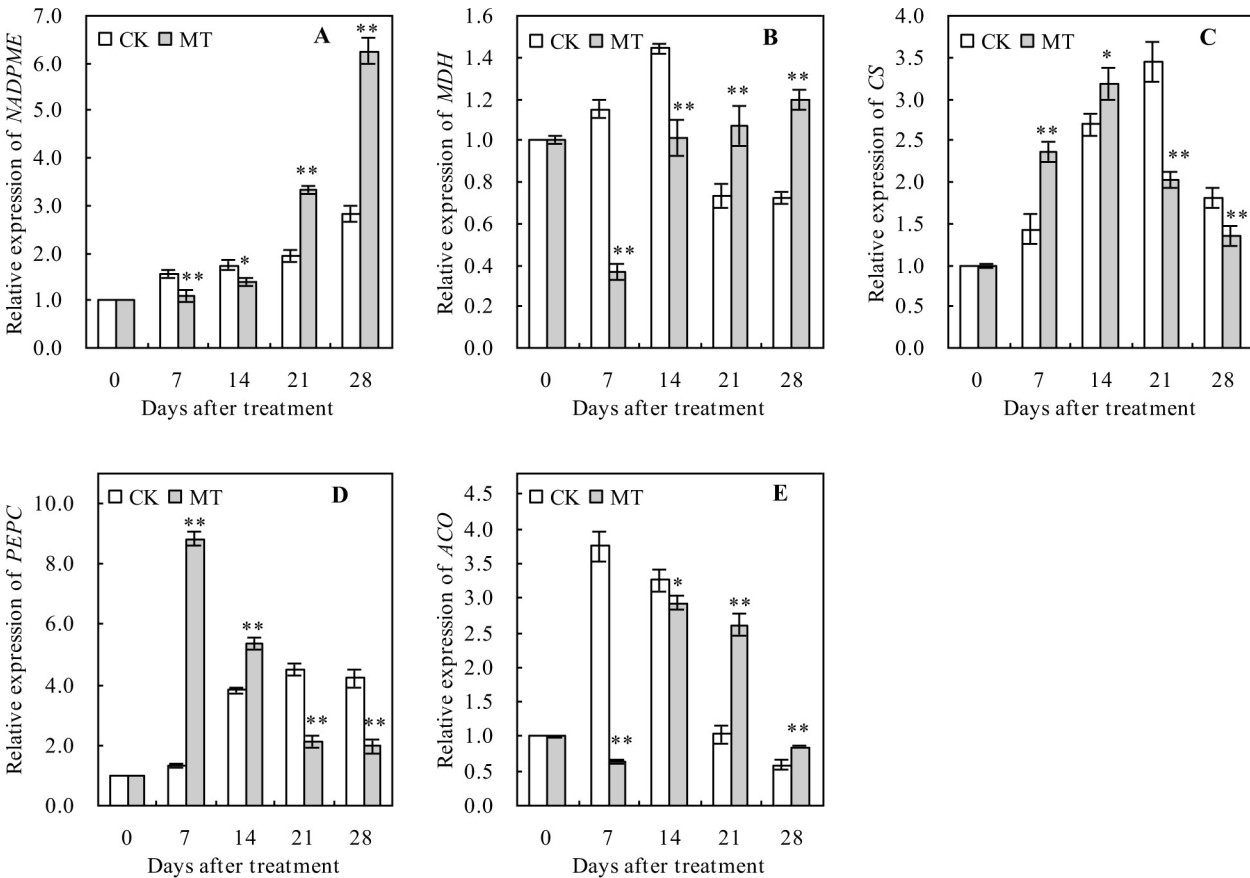

**Fig 6. Relative expression levels of organic acid metabolism-related genes in peach fruits.** Values represent the mean ± SD (n = 3). Asterisks indicate significant differences between the treatments using the Student's t-test (*: $0.01 \leq p < 0.05$; **: $p < 0.01$). Relative expression levels of **A:** *NADP+-malic enzyme* (*NADPME*), **B:** *malate dehydrogenase* (*MDH*), **C:** *citrate synthetase* (*CS*), **D:** *phosphoenolpyruvate carboxylase* (*PEPC*), and **E:** *aconitase* (*ACO*) genes.

activities to delay sucrose degradation in grape berries [39] and decreased SOX and SDH activities to delay sorbitol degradation in pear fruits [28]. In this study, MT treatment increased the SS, SPS, NI, and AI activities during the whole ripening period of peach fruits. This increased the accumulations of total sugar and sucrose at the later ripening stage of the fruits, which is related to the MT-mediated regulation of *SS*, *SPS*, *NI*, and *AI* gene expression levels [13, 40]. Therefore, MT treatment increased the relative expression levels of *SS*, *SPS*, and *NI* in peach fruits during the whole ripening period and increased that of *AI* at 14, 21, and 28 days after the treatment. This result further indicates that MT increases the SS, SPS, NI, and AI activities by regulating the gene expression levels of *SS*, *SPS*, *NI*, and *AI*. Moreover, the contents of total soluble sugar and sucrose increased in peach fruits with the increasing number of days after treatment, consistent with the previous studies [28, 38]. Therefore, MT treatment could increase the contents of total soluble sugar and sucrose in peach fruits by increasing the expression levels of their respective metabolism-related genes and enzyme activities during the whole ripening period. MT treatment also decreased the SOX activity of peach fruits during the whole ripening period but decreased that of SDH activity at 21 days, without significant effects at 7, 14, and 28 days after the treatment. These results are consistent with the previous studies on MT application on other fruit trees [28, 39] and related to the increased relative expression level of SDH in peach fruits 7, 14, and 21 days after treatment. The MT treatment only

increased the glucose and sorbitol contents of peach fruits at the mature stage and increased the fructose content 14 days after the treatment. These results are consistent with those previously observed in apricot fruits [41], which may be related to the increased SS and SPS activities and reduced SOX activity which mediates sucrose synthesis from glucose and fructose [40, 42]. However, the molecular mechanisms by which MT regulates sugar accumulation in peach fruits need further investigation.

Malic acid and citric acid are the main organic acids contributing to peach fruit flavor [38], with malic acid being the highest contributor [40]. These organic acids are interconverted through enzymatic catalysis, with ME oxidizing malic acid to pyruvate, while the phosphoenol-pyruvate (PEP), produced through sugar metabolism, is catalyzed by PEPC to oxaloacetate (OAA), which is then reduced to malic acid. MDH catalyzes the interconversion of OAA and citric acid, whereby OAA synthesizes citric acid through the action of citrate synthase (CS), while citric acid is converted to aconitic acid by aconitase (ACO). The MDH, CS, and PEPC are all known to promote the biosynthesis of malic and citric acids [43–46]. In this study, the contents of total organic acids and citric acid decreased in peach fruits, while that of malic acid increased at first and then decreased with the increasing number of days after the treatment. This result is consistent with those reported by the previous studies [41], suggesting that the contents of total organic acids, malic acid, and citric acid continuously decrease in peach fruits during the whole ripening period. During tomato fruit ripening, MT decreased the activities of PEPC and MDH, thereby reducing malic acid and citric acid accumulation [47, 48]. However, MT up-regulated the gene expression levels of *CS* and *ACO* to reduce the contents of malic acid and citric acid during orange fruit ripening [49]. In this study, MT treatment decreased total organic acids, malic acid, and citric acid in peach fruits at the mature stage. This result is consistent with the previous studies [47–49], suggesting that MT can also decrease the organic acid content of peach fruits. Additionally, MT treatment decreased CS and PEPC activities but increased the MDH activity in peach fruits at the mature stage, with no significant effects on the ME and ACO activities. MT treatment also decreased the relative expression levels of *NADPME*, *MDH*, and *ACO* and that of *CS* and *PEPC* at the early mature stage of peach fruits, while it had the opposite effects on these genes at the late mature stage of peach fruits. These results indicate that MT promotes the conversion of malic and citric acids to pyruvate and aconitic acids, thus reducing the contents of these organic acids. Therefore, MT can reduce the organic acid content of peach fruits by promoting the conversion of malic and citric acids; however, the molecular mechanisms by which MT regulates organic acid degradation in peach fruits need to be further investigated.

## Conclusions

This study demonstrated that spraying 150 μmol/L MT on the leaves of peach trees during the ripening period of peach fruits promoted the sugar accumulation in peach fruits by increasing the SS, SPS, NI, and AI activities and decreasing the SOX activity. Up-regulating the expression levels of *SS*, *SPS*, *NI*, and *AI* and down-regulating the expression levels of *SDH* also promoted sugar accumulation in peach fruits. MT also promoted organic acid degradation in peach fruits by increasing the MDH activity, decreasing the CS and PEPC activities, and up-regulating the relative expression levels of *NADPME*, *MDH*, and *ACO* but down-regulating the relative expression levels of *CS* and *PEPC*. However, further studies are necessary to understand the molecular mechanisms by which MT regulates sugar accumulation and organic acid degradation in peach fruits.

## Supporting information

**S1 Data.**
(XLS)

## Author Contributions

**Conceptualization:** Lijin Lin.

**Data curation:** Kexuan Zhou, Qi Cheng.

**Investigation:** Kexuan Zhou, Qi Cheng, Jingtong Dai, Yuan Liu, Qin Liu, Rui Li, Jiangyue Wang.

**Supervision:** Lijin Lin.

**Writing – original draft:** Kexuan Zhou, Qi Cheng.

**Writing – review & editing:** Rongping Hu, Lijin Lin.

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
