## [Decision Letter · Decision Letter 0]

7 Aug 2023

PONE-D-23-13723Effects of exogenous melatonin on sugar and organic acid metabolism in early-ripening peach fruitsPLOS ONE

Dear Dr. Lin,

Thank you for submitting your manuscript to PLOS ONE. After careful consideration, we feel that it has merit but does not fully meet PLOS ONE’s publication criteria as it currently stands. Therefore, we invite you to submit a revised version of the manuscript that addresses the points raised during the review process.

Please submit your revised manuscript by Sep 21 2023 11:59PM**.** If you will need more time than this to complete your revisions, please reply to this message or contact the journal office at plosone@plos.org. Please include the following items when submitting your revised manuscript:A rebuttal letter that responds to each point raised by the academic editor and reviewer(s). You should upload this letter as a separate file labeled 'Response to Reviewers'.A marked-up copy of your manuscript that highlights changes made to the original version. You should upload this as a separate file labeled 'Revised Manuscript with Track Changes'.An unmarked version of your revised paper without tracked changes. You should upload this as a separate file labeled 'Manuscript'.If applicable, we recommend that you deposit your laboratory protocols in protocols.io to enhance the reproducibility of your results. Protocols.io assigns your protocol its own identifier (DOI) so that it can be cited independently in the future. For instructions see: https://journals.plos.org/plosone/s/submission-guidelines#loc-laboratory-protocols. Additionally, PLOS ONE offers an option for publishing peer-reviewed Lab Protocol articles, which describe protocols hosted on protocols.io. Read more information on sharing protocols at https://plos.org/protocols?utm_medium=editorial-email&utm_source=authorletters&utm_campaign=protocols.

We look forward to receiving your revised manuscript.

Kind regards,

Bashir Sajo Mienda, PhD

Academic Editor

PLOS ONE

Journal Requirements:

"YES - This work was financially supported by the College Student Innovation Training Program Project of Sichuan Agricultural University (202110626079)."

"YES - This work was financially supported by the College Student Innovation Training Program Project of Sichuan Agricultural University (202110626079)."

Reviewers' comments:

Reviewer's Responses to Questions

**Comments to the Author**

1. Is the manuscript technically sound, and do the data support the conclusions?

Reviewer #1: Yes

2. Has the statistical analysis been performed appropriately and rigorously? 

Reviewer #1: Yes

3. Have the authors made all data underlying the findings in their manuscript fully available?

Reviewer #1: Yes

4. Is the manuscript presented in an intelligible fashion and written in standard English?

Reviewer #1: Yes

5. Review Comments to the Author

Reviewer #1: The research presented in the manuscript has investigated the effect of foliar supplied melatonin on metabolic changes in sugars and organic acids in peach fruit. Scientific claims made in the title, abstract and conclusion are justified with the graphical data presented in the manuscript. Organization, experimental design, and writeup of the manuscript is fairly good. Just a little note to provide full name of the abbreviation when used first time in the manuscript. In short, manuscript do not need major revision and is good enough to be considered for publication in the journal.

6. PLOS authors have the option to publish the peer review history of their article (what does this mean?). If published, this will include your full peer review and any attached files.

Reviewer #1: No

---

## [Author Response · Author response to Decision Letter 0]

29 Sep 2023

Reviewers' comments:

Reviewer's Responses to Questions

Comments to the Author

1. Is the manuscript technically sound, and do the data support the conclusions?

Reviewer #1: Yes

RESPONSE: Thank you for your reviewing.

2. Has the statistical analysis been performed appropriately and rigorously?

Reviewer #1: Yes

RESPONSE: Thank you for your reviewing.

3. Have the authors made all data underlying the findings in their manuscript fully available?

Reviewer #1: Yes

RESPONSE: Thank you for your reviewing.

4. Is the manuscript presented in an intelligible fashion and written in standard English?

Reviewer #1: Yes

RESPONSE: Thank you for your reviewing.

5. Review Comments to the Author

Reviewer #1: The research presented in the manuscript has investigated the effect of foliar supplied melatonin on metabolic changes in sugars and organic acids in peach fruit. Scientific claims made in the title, abstract and conclusion are justified with the graphical data presented in the manuscript. Organization, experimental design, and write up of the manuscript is fairly good. Just a little note to provide full name of the abbreviation when used first time in the manuscript. In short, manuscript do not need major revision and is good enough to be considered for publication in the journal.

RESPONSE: Thank you for your reviewing. We have revised for providing full name of the abbreviation when used first time.

6. PLOS authors have the option to publish the peer review history of their article (what does this mean?). If published, this will include your full peer review and any attached files.

Do you want your identity to be public for this peer review? For information about this choice, including consent withdrawal, please see our Privacy Policy.

Reviewer #1: No

RESPONSE: Thank you for your reviewing.

---

## [Editor Report · Decision Letter 1]

3 Oct 2023

Effects of exogenous melatonin on sugar and organic acid metabolism in early-ripening peach fruits

PONE-D-23-13723R1

Dear Dr. LIN,

We’re pleased to inform you that your manuscript has been judged scientifically suitable for publication and will be formally accepted for publication once it meets all outstanding technical requirements.

Kind regards,

Bashir Sajo Mienda, PhD

Academic Editor

PLOS ONE
---

## [Editor Report · Acceptance letter]

6 Oct 2023

PONE-D-23-13723R1 

Effects of exogenous melatonin on sugar and organic acid metabolism in early-ripening peach fruits 

Dear Dr. Lin:

I'm pleased to inform you that your manuscript has been deemed suitable for publication in PLOS ONE. Congratulations! Your manuscript is now with our production department. 

Kind regards, 

on behalf of

Dr. Bashir Sajo Mienda 

Academic Editor

PLOS ONE